# Genome-Wide Identification of the Aconitase Gene Family in Tomato (*Solanum lycopersicum*) and CRISPR-Based Functional Characterization of *SlACO2* on Male-Sterility

**DOI:** 10.3390/ijms232213963

**Published:** 2022-11-12

**Authors:** Zafer Secgin, Selman Uluisik, Kubilay Yıldırım, Mohamed Farah Abdulla, Karam Mostafa, Musa Kavas

**Affiliations:** 1Department of Agricultural Biotechnology, Faculty of Agriculture, Ondokuz Mayıs University, 55270 Samsun, Turkey; 2Burdur Food Agriculture and Livestock Vocational School, Burdur Mehmet Akif Ersoy University, 15030 Burdur, Turkey; 3Department of Molecular Biology and Genetics, Faculty of Science, Ondokuz Mayıs University, 55270 Samsun, Turkey; 4The Central Laboratory for Date Palm Research and Development, Agricultural Research Center (ARC), Giza 12619, Egypt

**Keywords:** aconitate hydratase, CRISPR/Cas9, male sterility, seedless tomato

## Abstract

Tomato (*Solanum lycopersicum*) is one of the most cultivated vegetables in the world due to its consumption in a large variety of raw, cooked, or processed foods. Tomato breeding and productivity highly depend on the use of hybrid seeds and their higher yield, environmental adaption, and disease tolerance. However, the emasculation procedure during hybridization raises tomato seed production costs and labor expenses. Using male sterility is an effective way to reduce the cost of hybrid seeds and ensure cultivar purity. Recent developments in CRISPR genome editing technology enabled tomato breeders to investigate the male sterility genes and to develop male-sterile tomato lines. In the current study, the tomato Acotinase (*SlACO*) gene family was investigated via in silico tools and functionally characterized with CRISPR/Cas9-mediated gene disruption. Genome-wide blast and HMM search represented two *SlACO* genes located on different tomato chromosomes. Both genes were estimated to have a segmental duplication in the tomato genome due to their identical motif and domain structure. One of these genes, *SlACO2*, showed a high expression profile in all generative cells of tomato. Therefore, the *SlACO2* gene was targeted with two different gRNA/Cas9 constructs to identify their functional role in tomatoes. The gene was mutated in a total of six genome-edited tomato lines, two of which were homozygous. Surprisingly, pollen viability was found to be extremely low in mutant plants compared to their wild-type (WT) counterparts. Likewise, the number of seeds per fruit also sharply decreased more than fivefold in mutant lines (10–12 seeds) compared to that in WT (67 seeds). The pollen shape, anther structures, and flower colors/shapes were not significantly varied between the mutant and WT tomatoes. The mutated lines were also subjected to salt and mannitol-mediated drought stress to test the effect of *SlACO2* on abiotic stress tolerance. The results of the study indicated that mutant tomatoes have higher tolerance with significantly lower MDA content under stress conditions. This is the first CRISPR-mediated characterization of *ACO* genes on pollen viability, seed formation, and abiotic stress tolerance in tomatoes.

## 1. Introduction

Tomato (*Solanum lycopersicum* L.) is one of the most produced fruits or vegetable crops that is produced five million ha all over the world. Tomatoes contain various health-promoting compounds essential to a balanced human diet. It is the natural source of vitamins C and E, as well as large amounts of carotenoids and polyphenolics, such as lycopene, sucrose, hexoses, citrate, and malate. Tomato is an indispensable component of meals in many cultures and one of the most important ingredients in the processed food industry [1]. In addition to their commercial and dietary importance, the tomato has become a model species for plant science due to its available genome sequence, ease of transformation, and short life cycle. It has been used to understand the molecular mechanism of fruit development and identify genetic responses to biotic and abiotic stress factors [2,3]. Since the discovery of heterosis (hybrid vigor) and the development of hybrid seed technology, most tomato varieties are grown from F1 hybrid seeds to have better offspring in terms of uniformity, yield, and stress tolerance. The common way to produce hybrid tomato seeds worldwide is the manual emasculation of male gametophytes to generate the female parent. However, this technology is a labor-intensive practice that significantly increases the cost of hybrid tomato seed production [4]. The failure of plants to produce functional anthers, pollen, or male gametes, which is also known as male sterility (MS), can be induced by genic or cytoplasmic systems, chemical hybridizing agents, or transgenic technology in plants. Incorporating male sterility into tomato breeding is a critical task to reduce the cost of hybrid seed production and ensure high varietal purity [5].

Over the years, researchers have developed several methods for integrating male sterility into the tomato genome. Several tomato genes controlling male sterility have been studied; many mutants, TILLING populations, recombinant inbred lines, and genetically modified tomato lines have been developed and tested in the last decades for this purpose [6,7]. Despite all this intensive research, superior male-sterile tomato lines have not been used on a large scale in tomato hybrid seed production. However, recent developments in genome editing technology have opened a new era to study gene function and develop new plant cultivars suitable for any condition. The latest version of the gene editing technology, CRISPR/Cas9, provides a novel and convenient method for producing male-sterile lines in soybean, maize, rice, wheat, and even tomato [8,9].

Aconitase, also known as citrate (isocitrate), hydrolyzes or aconitate hydratase (EC 4.2.1.3), is a metabolic enzyme containing Fe-S cluster that catalyzes the reversible isomerization of citrate to isocitrate via cis-aconitate in the tricarboxylic acid cycle (TCA). In humans, the *ACO1* gene is a cytosolic aconitase called iron-responsive element-binding protein 1 (IRP1) that contains an AcnA_IRP domain. IRP1 can recognize the RNA stem-loop structure of iron-responsive elements, localizes to the 5′ untranslated regions of ferritin mRNA, and inhibits ferritin translation [10]. Especially in higher plants, all aconitase sequences detected to date resemble cytosolic aconitase isoforms more than mitochondrial isoforms in animals [11]. Although the evolutionary patterns of the *ACO* family in land plants are poorly understood, the function of *ACOs* has been well-studied in model plants, such as Arabidopsis [12]. In Arabidopsis, inhibition of *AtACO* activity caused disturbances in the TCA cycle or cytosolic citrate metabolism [13]. Moreover, inhibition of aconitase by nitric oxide increased the expression of alternative oxidase, which is a regulator of mitochondrial stress responses [7]. Another study showed that the *OsACO1* gene in rice was induced by heat stress [14]. Transcriptome sequencing was performed in cytoplasmic male sterile lines of *Brassica napus*. Six genes involved in the TCA cycle, mainly encoded aconitase, were shown to be significantly down-regulated [15]. Although aconitase genes are expressed at low levels during most developmental stages, a dramatic increase in its expression is observed during seed, pollen maturation, and germination [16]. Northern blot analysis was used to examine the expression of three *AtACO* genes in different plant organs, including the seed, root, stem, leaf, and flower. [17]. In the same study, the germination rate of seeds obtained by crossing *ACO* mutant Arabidopsis with non-mutant lines was reduced by 23–25%. The double mutant, *Ataco1*/*Ataco3*, caused abortion in seeds, implying that the two genes have a crucial role in embryo formation and seed development. Based on these studies, the relationship of the *ACO* genes with citric acid contents, oxidative stress, and iron homeostasis was determined in plants. However, to the best of our knowledge, there is no study characterizing an *ACO* gene related to pollen development and male sterility in plants. Therefore, the current study aimed to identify *SlACO* genes in tomato genome using bioinformatics tools. Then, CRISPR/Cas9-mediated loss-of-function assay was carried out in a mutant tomato line to investigate the potential regulatory role of *SlACO2* (cytoplasmic) genes involved in tomato male sterility and seed formation.

## 2. Results

Putative *aconitase (SlACO)* genes were searched against *Solanum lycopersicum* genome using various in silico tools, and two genes were identified (*SlACO1-Solyc07g052350.3.1* and *SlACO2-Solyc12g005860.2.1*). Detailed information on each gene is listed in Table 1. During this analysis, two more genes (*Solyc03g005730.4.1* and *Solyc09g090900.4.1*) containing the aconitase domain but encoding the enzyme 3-isopropyl malate dehydratase were identified in the tomato genome. Since these genes do not have aconitase and swivel domains together, they were not considered true aconitase genes and were not included in subsequent studies. Both *SlACO* proteins contain an aconitase and swivel domain, and their protein lengths range from 982 to 996 amino acid residues. Their molecular weights (Mol. Wt) were found to be 107.15 kDa and 108.2 kDa, respectively. Sequence analysis of *SlACO1* and *SlACO2* showed that the isoelectric point (pI) of the *SlACO* proteins was 6.52 and 7.29, respectively, and the large mean hydropathicity (GRAVY) was −0.178 and −0.223. The results of the instability index show that *SlACO*s are stable in the test tube. (instability index > 40). The predicted subcellular localization of *SlACO* proteins indicated that these proteins were highly active in plastid and mitochondria (Appendix A). Examining the exon-intron structure of each member of *SlACO* enabled us to discover more about the structural variation among *SlACO* genes. As illustrated in Figure 1c, both *SlACO1* and *SlACO2* genes have 20 exons. In addition, the intron length of *SlACO* genes was observed to be variable in tomato plants. It is generally agreed that protein motifs play a vital role in the interaction of different modules in transcriptional complexes and the transmission of signals and that they are also intimately related to gene classification [18]. In this way, the phylogenetic relationship and classification of *SlACO* genes were supported by motif analysis. Six conserved motifs of *SlACO* proteins were analyzed using the MEME suite (Figure 1a). Chromosomal locations of the *SlACO* genes were visualized using TBtools software, and it was found that *SlACO1* and *SlACO2* were located on chromosomes 7 and 12, respectively. Gene duplication events determined that there are segmental duplications between the *SlACO1*/*SlACO2* (Appendix A). The synonymous ratio (Ks), non-synonymous ratio (Ka), and Ka/Ks of these genes were calculated, and the results are given in Appendix A.

In order to understand the expansion of aconitase hydratase genes in different plant species, the full-length amino acid sequences of 28 *ACO* genes from six model plants were used to construct the phylogenetic tree using the maximum likelihood method (Appendix A). A total of 26 aconitase genes, three in *Arabidopsis thaliana* (*AtACO1*, *AtACO2*, and *AtACO3*), seven in *Zea mays*, six in *Medicago* truncatula, four in *Populus trichocarpa*, four in *Sorghum bicolor*, and two in *Brachypodium* distachyon genome, were identified to be used in phylogenetic analysis. The result showed that the 28 *ACO* genes were divided into ten subgroups, here named groups I to X. *SlACO* genes are clustered in III subgroups. In this phylogenetic classification, subgroups I, II, and X are represented by one member each, and subgroups III, IV, V, and IX are represented by two members, but most are in subgroup VII. In order to figure out conserved amino acids in the swivel domain of *SlACO*s and *AtACO*s, a ClustalW alignment was performed. In the analysis performed to compare the two organisms in terms of the swivel domain consisting of 131 amino acids, only 28 amino acid differences were found; that is, 78.6% of them were completely conserved (Appendix A).

### 2.1. In Silico Functional and Expressional Characterization of Tomato SlACO Genes and Determination of the SlACO2 Expression Level by RT-qPCR

To study the tissue-specific expression patterns of members of the *SlACO* gene family, we used pollen cell development-related RNA-seq data to examine functional gene expression analysis in tomatoes [19]. As shown in Appendix A, *SlACO2* was highly expressed in microspores, mature pollen cells, and generative and sperm cells examined in the study. Among these four tissues that were used for the analysis, it was discovered that the expression of the *SlACO1* gene was only down-regulated in the generative cell, while it was up-regulated in all of the other tissues. According to the results obtained from Tomato eFP browsers, it has been determined that both *ACO* genes have high activity in all tissues such as flower, vegetative meristem, seedling root, seedling shoot, developing fruit, mature fruit, leaves, stem (Appendix A).

The highly correlated 25 genes related to pollen development were extracted based on the publicly available transcriptome data (GSE109672 and GSE117185). In these studies, pollen development at tetrad, post-meiotic and mature stages were investigated in different tissues, including microspores, generative cells, pollen cells, callus, fully expanded leaves, pistil, and mature pollen grain (Appendix A). *SlACO* genes, four genes related to male sterility; *SlPHD_MS1* [20], *SlMS10* [21], *SlAMS* [5], *SlSTR1* [22], and these 25 genes were hierarchically clustered by Manhattan distance in three anatomical parts including microspores, generative cell and pollen cell (Appendix A). Three sub-clusters were observed, and the *SlACO2* gene was found to cluster separately from *SlACO1*. Later, the STRING program was used to understand the functional interaction between those 25 genes and two genes related to male sterility with *SlACO*s (Appendix A). Four gene clusters were created with 0.625 local clustering coefficient score based on the MCL inflation parameter (Appendix A). Ten predicted functional partners were revealed to create the functional interaction network of *SlACO*s with pollen development and male sterility (Appendix A). There are 31 nodes and 99 edges. Protein-protein interaction enrichment *p*-value is less than 1.0e-16. All of the findings that were obtained indicate that it is reasonable to choose *SIACO2* as the target in plants to be created with the CRISPR/Cas9 for the purpose of performing functional characterization of the process of pollen formation.

### 2.2. Generation of SlACO2 Mutants Using the CRISPR/Cas9 Gene-Editing System

In order to investigate the male-sterility function of *SlACO2*, mutants were generated using the CRISPR/Cas9 gene editing technology. Two target sites, Target 1 (Exon 1) and Target 2 (Exon 2) were selected for *SlACO2* knock-out studies (Figure 2a). Both gRNAs were separately cloned into the plant expression vector pKI1.1R and expressed under the control Arabidopsis constitutive U6-26 promoter (Figure 2b). The resulting constructs were introduced into Agrobacterium tumefaciens strain GV3101, and the tomato cultivar ‘Crocker’ was used for transformation separately. A total of 50 cotyledon explants were used for each gRNA. Callus formation started after 10–12 days of transformation, and putative genome-edited plants were formed after 30 days. A total of 22 independent transgenic plants, 18 of which were from gRNA1, were obtained by Agrobacterium-mediated transformation. Using PCR and gene-specific primers for the *hptII* gene, we could validate the presence of transgenes (Appendix A).

Sanger sequencing was used to check for mutations in the PCR product that was produced from four lines of gRNA2 and six lines of gRNA1. As a result of Blast and Synthego ICE analyses, it was determined that gRNA2 did not cause any mutations, and Line 1, line 2, line 4, and line 5 plants had the same mutation pattern in gRNA1. Therefore, considering these lines may be clones, only the results of line 1, line 3, and line 6 were given as a result of Sanger sequence analysis (Figure 3). As seen in Figure 3, although there is a homozygous mutation in line 3 and line 6, a heterozygous mutation is observed in line 1. In this context, 13 base deletions occurred in line 3 and two base deletions in line 6.

### 2.3. Control of Pollen Viability and Determination of Seed Formation in SlACO2 Mutant Lines

The TTC staining method was applied to determine the pollen viability of the *SlACO2* mutant induced by gRNA1 and WT tomato plants. About 85% of pollen was positively stained red in WT plants, indicating viability, whereas 4% and 2% of the pollen grains were stained in Line 3 and Line 6 mutant lines, respectively (Appendix A, Figure 4). However, no difference was observed in WT and mutant lines in terms of flower structure and flower color (Figure 4).

Pollen and anthers from wild-type (WT) and mutant Line 3 plants were compared using scanning electron microscopy to determine if silencing of *ACO* genes results in a morphological difference in mutant plants. As seen in Appendix A, there is no difference between the pollen grains of the plants compared in terms of size and shape. Although many pollen grains were observed in the anther tissue of the WT plant, less pollen formation was observed in the line 3 plant.

Once the ripening process was finished, and the development of the seeds was brought under control, the tomatoes were picked. The morphologies of wild-type and mutant plants’ fruits were compared, as well as the number of chambers and clusters that formed within the fruits of each of the two plants (Figure 5). According to this information, no discernible difference was found between the WT and *SlACO2* mutant lines (Line 3 and Line 6) regarding the number of clusters counted as seven and the number of fruit chambers counted as two. The number of seeds per fruit was also determined by calculating the average seed number of 10 fruits in WT, Line 3, and Line 6 mutant lines. The results showed that the number of seeds/fruit of WT plants was 67; it was observed as 10.2 in the Line 6 plants indicating that 85% of seed formation was inhibited in mutant lines compared to WT plants (Appendix A). Although there is a fairly big difference in the total quantity of seeds produced by wild-type and mutant plants, the seeds produced by either kind do not differ in terms of their size or shape (Appendix A).

### 2.4. Expression Analysis of SlACO2

At different developmental stages, expression levels of the *SlACO2* gene in various tissues, including flower, bud, and leaf, were evaluated using qRT-PCR in both WT and mutant plants to confirm the effectiveness of the CRPSR/Cas9 system (Figure 6a). It was found that the expression of the silenced *SlACO2* gene in all tissues at different developmental stages analyzed was significantly decreased in genome-edited plants compared to WT plants (*p* < 0.01). This decrease was more in line 3 plants. The highest decrease was observed in the 2nd and 3rd stage flower tissues when WT was compared with the plant (Figure 6b). According to the WT plant, the tissue in which the target gene works most is the 3rd stage leaf tissue isolated from Line 6.

### 2.5. Evaluation of the Effect of Salt and Drought Stress on Genome-Edited Lines

To simulate salinity stress, we germinated seedlings of WT and *SlACO* mutant lines on MS media supplemented with 200 mM NaCl under in vitro conditions. In the same manner, 200 mM mannitol was added independently to simulate drought stress. Leaves of 10-week-old seedlings were collected for MDA and proline analysis. As shown in Figure 7, there was no significant difference in the level of proline content on tomato seedlings under control conditions. Line 6 showed a significantly lower proline content when treated with NaCl as well as mannitol treatment compared to WT. On the contrary, no significant difference was observed in the proline content comparing line 3 to WT in both treatments. A significantly lower level of MDA content was noticed under control treatment on both line 3 and line 6 compared to WT. Similarly, line 3 and line 6 mutants exhibited a significantly lower MDA content compared to that of WT when treated with NaCl and mannitol.

## 3. Discussion

Compared to cytoplasmic male sterile lines, it is much easier and more useful to produce hybrid seeds with the use of genome editing techniques [23]. Thus, the development of germplasm sources of male sterile lines depends on developing novel crop production strategies. The male sterile phenotype is usually controlled by a recessive mutation of the nuclear gene; therefore, it is possible to generate male sterile mutants using the CRISPR/Cas9 system. The *SlMS10* gene, which encodes the basic helix-loop-helix transcription factor (bHLH), is involved in meiosis and programmed cell death in the tapetum during microsporogenesis. Therefore, modification of the SlMS10 via CRISPR/Cas9 leads to male sterility in tomato plants [21]. In a recent study using AMS, another gene encoding a basic helix-loop-helix (bHLH) TF was knocked out by CRISPR/Cas9 and over-expressed in tomatoes. SlAMS downregulation or upregulation led to abnormal pollen development, which in turn decreased pollen viability, and subsequently generated male-sterile lines [5]. In our laboratory, we have generated completely male-sterile lines by mutating the *SlPHD_MS1* gene (*Solyc04g008420*), encoding a *PHD*-type transcription factor [20]. In another study, male-sterile lines were obtained by silencing the strictosidine synthase-like (*SlSTR1*) gene, a homolog of the *Ms45* gene of the maize plant in tomato [22]. Taken together, these results demonstrate the significance of a biotechnology-based genetic male-sterility system, including the development of novel male-sterile tomato lines.

In this study, in vitro and in silico research methods were utilized to examine the aconitase gene found in the tomato genome. In keeping with the literature, such as the information that there are three *ACO* genes in both Arabidopsis [13] and Citrus [24], two *ACO* genes were found in the tomato genome. Phylogenetic analysis showed that two *SlACO* proteins, *SlACO1* (Solyc07g052350) and *SlACO2* (Solyc12g005860), fall within the same clade, which has the aconitase swivel superfamily domain in addition to aconitase superfamily domain (Appendix A). Moreover, it was reported that these two genes participate in the functioning and development of male gametophytes by having a role in energy-related pathways (Glycolysis and Krebs cycle) as a prerequisite for pollen maturation [25]. Additionally, the expression pattern of the *SlACO*s is presented in diverse tissues of tomato, including microspores, mature pollen, and generative cell (Appendix A), in which *SlACO2* had the strongest expression in all tissues. Collectively, this evidence implies that *SlACO2* is a possible target for editing to create male sterility in tomatoes.

Gene editing technologies are becoming powerful tools for crop improvement and breeding. Among them, the CRISPR/Cas9 system has been extensively used for genome editing in various organisms, especially plants, due to its simplicity, high efficiency, and multiplex capabilities. CRISPR/Cas9 technology was used in this study to knock out *SlACO2* to confirm the results of in silico tools. The TCA cycle, including the aconitase enzyme, has significant application potential in improving crop yields [4]. There have been numerous studies showing that TCA cycle-related genes involved in seed germination [26], root development [27], flower development [28], and fruit ripening [29]. Aconitase is an iron-sulfur (Fe-S) enzyme involved in the assembly of a very complex cluster. *Aco1*/*Aco3* double mutant of Arabidopsis caused the abortion of seeds, indicating the two genes had a crucial role in embryo formation and proper seed development [17]. Here, we generated six *SlACO2* mutant tomato lines and confirmed the mutagenesis using molecular and physiological analysis. While two of them were found to be homozygous, the rest were heterozygous. Even though the flowers and seeds generated by homozygous mutant lines seemed identical to those produced by wild-type lines, homozygous mutant lines produced an extremely low number of seeds. A finding that scanning electron microscope analysis of mutant flower anthers reveals they contain very little pollen contributed to the credibility of this evidence. Additionally, the results of our study do not appear to be consistent with the information found in the previous research by Carrari, and Nunes-Nesi [30]) in which the tomato plants created by silencing of the *ACO1* gene produced darker leaves, indicating a greater quantity of carotenoids. On the contrary, we found that the leaves of wild-type plants and mutant lines were indistinguishable from one another.

To further characterize the remarkable phenotypic difference between WT and *SlACO* mutant lines, physiological studies were applied by treating tomato seeds with abiotic stressors in the form of salt and drought. The level of MDA content is used as a direct marker of membrane damage and lipid peroxidation to measure cell damage in plants under environmental stress [31]. In our study, mutant lines showed a significant reduction in MDA content compared to WT, indicating reduced oxidative damage caused by salinity and drought stress. This result is in accordance with previous reports for the level of MDA content as inversely proportional to the degree of tolerance to abiotic stress [32,33,34,35]. An increase in proline content in cells can function as an osmotic adjustment mediator, an eliminator of free radicals, and a potential redox buffer under environmental stress [36,37]. In this study, no significant difference in proline content was observed in Line 3 as opposed to an increase in the proline level in Line 6 compared with WT. In line 6, the level of free proline content was significantly reduced. This underlined the fact that *SlACO2* mutant line 6 might augment drought and salinity stress-induced membrane damage. A similar result was previously reported in tomato plants with reduced proline accumulation [34,35].

This suggests that the loss of function of *SlACO* might be related to lower ROS production, which can be linked to enhanced resistance to oxidative damage and increased tolerance of tomato seedlings to salt and drought stress.

## 4. Materials and Methods

### 4.1. Genome-Wide Identification and Analysis of SlACO Genes

The *AtACO1* sequence (AT4G35830) was obtained from the NCBI (https://www.ncbi.nlm.nih.gov/gene/, accessed on 10 October 2022) and used as a query for blast search against the tomato genome in the Phytozome v13 (https://phytozome-next.jgi.doe.gov/, accessed on 10 October 2022) database to identify probable *SlACO* genes. After blast analysis, a sequence search was performed using the Pfam database (http://pfam.xfam.org/, accessed on 10 October 2022) to find the domain sequence of this gene with a length of 795 amino acids. Putative *SlACO* genes were identified by searching for proteins containing the Aconitase domain using Hidden Markov Model (HMM) search tool. The genes containing the Aconitase domain from the tomato plant were analyzed using the HMMER 3.1b2 database (http://hmmer.org, accessed on 10 October 2022). The length of the open reading frame (ORF), the isoelectric point of proteins (pI), and the molecular weight (MW) of identified *ACO* proteins were analyzed using the ProtParam tool on the ExPASy web server (http://www.expasy.ch/tools/pi_tool.html, accessed on 10 October 2022) and Phytozome v13 web tool (https://phytozome.jgi.doe.gov/pz/portal.html, accessed on 10 October 2022).

### 4.2. Phylogenetic Analysis and Multiple Sequence Alignment of SlACO Genes

Using the same methodology that was utilized for the identification of *SlACO*s, the full-length amino acid sequences of *ACO* proteins derived from *Arabidopsis thaliana*, *Populus trichocarpa*, *Medicago truncatula*, *Zea mays*, *Brachypodium distachyon*, and *Sorghum bicolor* were determined. Alignments of amino acid sequences of *ACO* proteins were performed with ClustalW, and the phylogenetic tree was generated with MEGA X software using the neighbor-joining (NJ) method with bootstrap analysis (1000 replicates) [38].

### 4.3. Exon/intron Structure and Conserved Motif Analysis of SlACO Genes

The DNA, protein, and CDS sequences of *SlACO*s were downloaded from the Phytozome v13 database. Conserved motif analysis of *SlACO* protein sequences was discovered with an online MEME (https://meme-suite.org/meme/, accessed on 10 October 2022) server and visualized with TBtools software [39]. In the process, the optimum width was set to 80–100 amino acids, and any number of motif repetitions and the maximum number of motifs was selected as 10.

### 4.4. Chromosome Location and Determination of Gene Duplications of SlACO

Distributions of *SlACO* genes on chromosomes and gene duplications were performed using TBtools software (https://github.com/CJ-Chen/TBtools, accessed on 10 October 2022). The tomato-tomato block file was downloaded using the Plant Genome Duplication Database (http://chibba.agtec.uga.edu/duplication/, accessed on 10 October 2022) to determine gene duplications. Ka/Ks ratios were calculated using TBtools software by selecting the data of *SlACO* genes from the obtained Block file.

### 4.5. In Silico Expression Analysis of Tomato SlACO Genes

Expression patterns of *SlACO* genes were examined using transcriptome data of the developmental profile of tomato pollen cells obtained from the NCBI GEO dataset (GSE117185) [19]. Expression profiles of *SlACO* genes were visualized with TBtoools using transcriptome data. Putative tissue-specific expression profiles of *SlACO1* and *SLACO2* genes were extracted based on the Solanum lycopersicum transcript expression database from several tissues and organs, including flowers, leaves, roots, and fruit from different developmental stages. Expression profiles were built using the tomato plant Electronic Fluorescent Pictograph Browsers (Tomato eFP browsers) (http://bar.utoronto.ca/eplant_tomato/, accessed on 10 October 2022). We predicted the subcellular localization of SLACO proteins on tomatoes using an online TargetP-2.0 server (https://services.healthtech.dtu.dk/service.php?TargetP-2.0, accessed on 10 October 2022) [40].

### 4.6. Construction of Cas9/gRNA-Expressing Vectors

The gRNAs targeting *SlACO2* were designed with the help of CRISPR-P V2.0 (http://crispr.hzau.edu.cn/CRISPR2, accessed on 10 October 2022) tools. The target site was selected based on criteria, including on-score value ((best score > 0.50) and moderate (0.20 < score < 0.50)), GC content (30–80%), number of complementary bases (TBP) ≤ 12 between gRNA and gRNA scaffold sequences and off-target capacity to efficiently disrupt gene function. Features of candidate gRNAs are provided (Figure 8). Each of the two synthesized gRNAs was cloned separately into the pKI1.1R (Addgene plasmid #85808) plasmid.

### 4.7. Generation of Transgenic CRISPR/Cas9 Mutant Lines in Tomato

The Cas9/gRNA constructs harboring the gRNA1 and gRNA2 targeting the *SlACO2* gene were transformed into Agrobacterium tumefaciens strain Gv3101 using electroporation separately. Agrobacterium-mediated transformation of tomato (Solanum lycopersicum) cultivar ‘Crocker’ were performed according to Secgin et al. (2021) [41]. All tomato lines were grown in standard glasshouse conditions of 16-h day length and 24–26 °C, with a night temperature of 18 °C. Plants from each line were grown in coarse potting compost (Klasmann Potgrond H) in 30 L pots with standard Hoagland’s solution irrigation treatment. Tomato fruits were harvested at commercial maturity and used for further analysis.

### 4.8. Identification of Transgenes and CRISPR/Cas9 Mutation

Genomic DNAs from the leaves of transgenic tomato and WT plants were isolated using the Quick-DNA™ Plant/Seed Miniprep Kit (Zymo Research, Orange, CA, USA) to evaluate the mutation of target sites. The first PCR reaction was carried out to confirm the presence of the transgene in T0 plants using *HptII* gene-specific primers (Table 2). Then, specific PCR primers were utilized to amplify the genomic regions spanning the target *SlACO2* sites, which were subsequently examined to look for CRISPR/Cas9-induced mutations. Sanger sequencing was applied to find possible mutations in amplified PCR products. The obtained sequences were compared with the wild-type reference sequence of the *SlACO2* using BLASTN NCBI. By using the Synthego ICE CRISPR Analysis Tool [42] and DSDecodeM [43], the mutation types and their rates were examined. In addition to the target gene, a potential off-target gene (*Solyc01g014620.1*) having four mismatches was also sequenced.

### 4.9. Characterizing the Morphology of Mutant Plants

The morphological and structural properties of T0 and WT mutant lines’ flowers and pollens were studied. Triphenyl tetrazolium chloride (TTC) was used in an assay to measure pollen vitality. The slide was covered with a layer of pollen taken from various plant lines. The pollen was left exposed to TTC for two hours before being placed in the dark. Pollen was counted using a Nikon Eclipse E200 microscope equipped with 10 × 20 oculars and a 40/0.25 objective on samples stored in the dark. The TTC test classified undyed pollen as dead, partially active pollen as pink, and fully active pollen as red. Additionally, pollen grains and anthers of genome-edited and WT plants were visualized using scanning electron microscopy (JEOL JSM-7001F). Following this analysis, the structural makeup of flowers from various plant strains was analyzed. The physical characteristics of the flowers were compared, including their size, stigma, and stamen structure.

### 4.10. Physiological Analysis of Genome-Edited Plants

For the functional analysis of the *SlACO* transgenic line against WT related to abiotic stress, T1 generation seeds of WT and transgenic lines were firstly surface sterilized using 5% sodium hypochlorite, as mentioned above. Surface sterilized seeds were then cultured on MS medium supplemented with either 200 mM NaCl to simulate salinity stress or 200 mM mannitol to simulate water deficit. Seedlings were then allowed to germinate in growth chambers set constant at 24 ± 2 °C with a 16-h light and 8-h dark photoperiod at a light intensity of 400 μmol m^−2^ s^−1^. Treatment was carried out on glass jars containing five seedlings each and three jars representing three biological replicates per treatment. Leaf samples from 10-week-old seedlings were selected randomly for physiological assay, including malondialdehyde (MDA) and proline contents, and immediately frozen in liquid nitrogen and then stored at −80 °C.

As previously described, the MDA level of transgenic lines and WT was determined using the thiobarbituric acid method [44]. In brief, 200 mg leaf samples of control, NaCl, and mannitol-treated seedlings were homogenized with 1ml 0.5% trichloroacetic acid (TCA). The total amount of ground homogenates were then centrifuged for 15 min at 12,000× *g* at 4 °C. A 0.6 mL volume of supernatant was mixed with the same volume of 0.5% TBA containing 5% TCA in a single Eppendorf tube. The mixture is then boiled for 25 min at 96 °C. After cooling the mixture on ice to room temperature, the mixture is centrifuged for an additional 5 min at 10,000× *g*. Absorbance was then recorded at 532 nm and 600 nm using 0.5% TBA containing 5% TCA as blank.

Proline content was extracted and determined using the acid ninhydrin method as previously described [45]. 200 mg of frozen leaf samples were homogenized with 1 mL of 3% sulfosalicylic acid. The homogenized extract is then centrifuged for 5 min at 14,000× *g* at 4 °C. The supernatant (0.1 mL), acid ninhydrin (0.2 mL), 96% acetic acid (0.2 mL), and 3% sulfosalicylic acid (0.1 mL) are mixed in a single Eppendorf tube and incubated on 95 °C hot blocks for one hour. After the mixture is cooled down on the ice, 1 mL toluene is added and then centrifuged at 14,000× *g* for 5 min at 4 °C. The absorbance of the red supernatant was then read at 520 nm, and a preset calibration curve was used to record the proline content. All experiments were performed with triplet replicates, and results were presented with standard deviations and standard errors computed using IBM SPSS 26 software. A one-way ANOVA analysis was used to analyze the data, and a difference of *p* < 0.05 (*) was considered statistically significant, and *p* < 0.01 (**) was considered very significant compared with WT.

### 4.11. Total RNA Isolation and qRT-PCR Analysis

qRT-PCR was performed to determine the gene expression of the *SlACO2* at different developmental stages of flowers and different tissues of mutant and WT tomato plants. Total RNA was isolated from various tissues of tomato plants using QIAGEN RNeasy Plant Mini Kit (QIAGEN, Valencia, CA, USA) following the manufacturer’s protocol. RNA concentration was quantified spectrophotometrically. According to the manufacturer’s instructions, one µg of RNA was reverse transcribed to cDNA using the iScript™ cDNA Synthesis Kit (Biorad). All reactions were performed with a CFX96 (Biorad) following the protocol; 98 °C for 30 s, followed by 40 cycles of 95 °C for 15 s, and 60 °C for 30 s. For each sample, three biological replicates and two technical replicates were analyzed. The actin one gene was used as an internal reference gene to obtain relative transcript abundances for SYBR green-based real-time PCR assay. The relative gene expression was calculated using the 2^−ΔΔCT^ method [46], and the primer sequences used for qRT-PCR are listed (Table 2). Data provided represents fold expression Log2 (WT/genome-edited lines). The one-way ANOVA was used to determine the statistical significance of changes in relative *SlACO2* expression levels. The means and standard deviation of the replications were compared by the least significant difference (LSD) test at *p* ≤ 0.01.

## 5. Conclusions

In this study, we identified two aconitase genes in the tomato genome, which we think are associated with male sterility. In silico and in vitro expression analyses of these genes were performed. The *SlACO2* gene, which was determined to be highly expressed in different generative tissues, was knocked out using the CRISPR/Cas9 system. Sequence analysis of six mutant lines obtained using gRNA1 was performed, and Lines 3 and 6 were determined to be homozygous mutants. In the morphological analysis, it was determined that there was no difference between mutant and WT plants. On the contrary, in physiological analysis, it was determined that the MDA content of mutant plants decreased significantly compared to WT plants, and mutant plants were less affected by applied stresses.

## Figures and Tables

**Figure 1 ijms-23-13963-f001:**
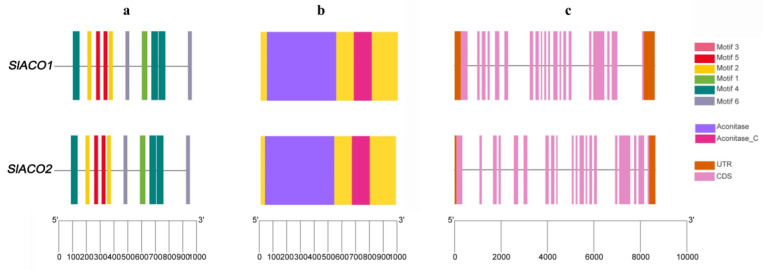
Motif structure, conserved domains, and exon-intron architecture of *SlACO* genes. (**a**) Structure of 6 conserved motifs analyzed through MEME suite tool online. Legends indicate the color of each motif. (**b**) Conserved domain structure of *SlACO* genes in tomato. Two conserved domains were found and presented in purple and dark pink colors. (**c**) Exon-intron architecture of *SlACO* genes. The light pink boxes indicate exons, the solid line indicates introns, and the brown boxes indicate the non-coding untranslated regions.

**Figure 2 ijms-23-13963-f002:**
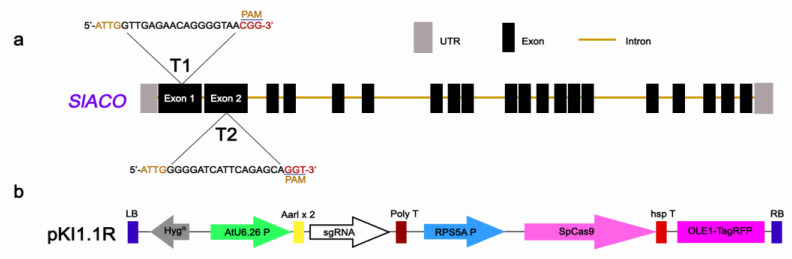
CRISPR/Cas9-mediated genome editing of *SlACO2* gene (**a**) Schematic illustration of the two target sites in *SlACO2* genomic sequence. Target 1 and Target 2 sequences are shown in capital letters, and the protospacer adjacent motif (PAM) sequence is marked in red. (**b**) Schematic diagram of Cas9/sgRNAs expression vector (pKI1.1R) used for Agrobacterium-mediated transformation of tomato.

**Figure 3 ijms-23-13963-f003:**
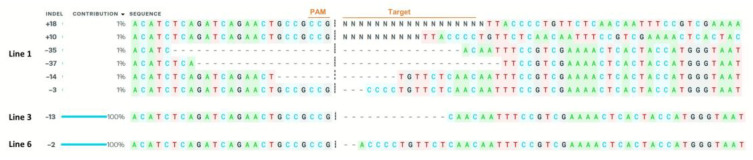
Analysis of CRISPR/Cas9-induced knock-out lines using ICE CRISPR Analysis Tool. Deletions are indicated by a dotted line. The PAM site is shown in red.

**Figure 4 ijms-23-13963-f004:**
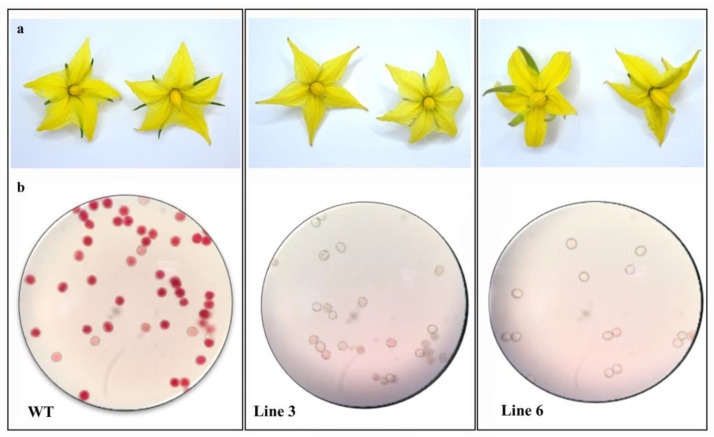
Morphological comparisons of WT and *SlACO2* mutant plants. (**a**) Flower structures of WT and *SlACO2* mutant plants. (**b**) Light microscopy photos of pollen viability in WT and mutant plants (Line 3 and Line 6) obtained by TTC test.

**Figure 5 ijms-23-13963-f005:**
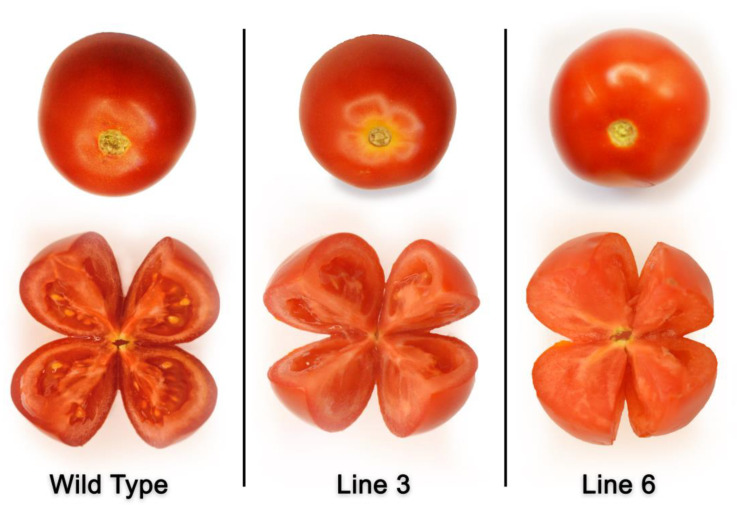
Harvested fruits and seeds formed in ripe fruits of WT and *SlACO2* mutant plants.

**Figure 6 ijms-23-13963-f006:**
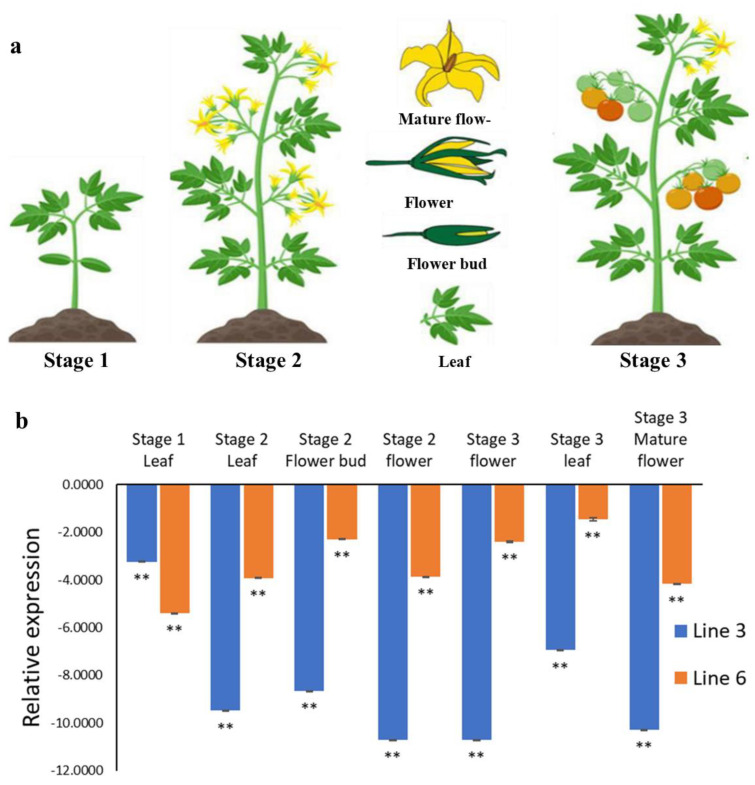
Relative gene expression between WT and genome-edited lines at different stages. (**a**) Explants and their stages used for qRT-PCR analysis. (**b**) Relative expression analysis of *SlACO2* knockout and WT plants by qRT-PCR at different developmental stages of leaf and flower tissues (the Actin gene was used as an internal reference gene). Data provided represents fold expression Log2 (WT/genome-edited lines). Values represent the average ± SD of three biological replicates of each reaction. ** Indicate significant differences at *p* < 0.01.

**Figure 7 ijms-23-13963-f007:**
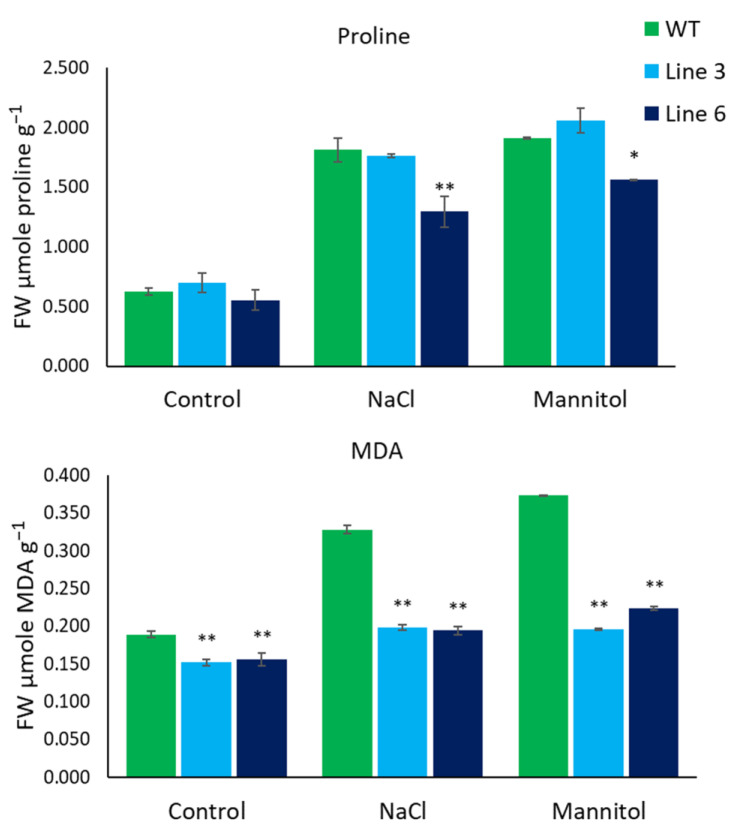
Comparative analysis of proline and MDA content in WT and *SIACO2* mutant lines under control, 200 mM NaCl, and 200 mM mannitol treatments from 10 weeks old leaves. Results represent mean values; error bars indicate the ± standard error (n = 3 replicates). Single asterisks indicate statistically significant differences (*p* < 0.05), and double asterisks represent a very significant difference (*p* < 0.01) according to ANOVA (IBM, SPSS 26).

**Figure 8 ijms-23-13963-f008:**
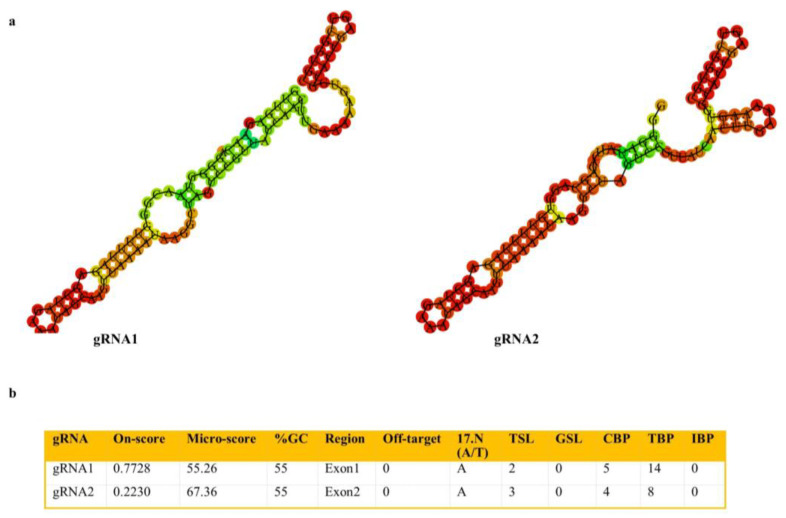
Some features and selection criteria of designed gRNAs targeting *SlACO2*. (**a**) Minimum free energy structures of gRNAs and scaffold estimated by RNA Fold Server. (**b**) Properties of designed gRNAs provided by Crispr-P-V2.0.

**Table 1 ijms-23-13963-t001:** Physiochemical properties of *SlACO* proteins.

Gene Name	Chromosome	Length (bp)	CDS (pb)	Protein Length (A.A)	Accession Number	pI	Molecular Weight (Kda)	Instability Index (II)	Stability	GRAVY	Solubility/Score
*SlACO1*	ch07	8621	2988	996	XP_004243472.1	7.29	108.20	32.74	stable	−0.178	Soluble/0.552
*SlACO2*	ch12	8650	2946	982	XP_004251517.2	6.52	107.15	35.57	stable	−0.223	Soluble/0.568

**Table 2 ijms-23-13963-t002:** List of primers used in this study.

Primer Name	Sequence 5′-3′	PCR Product
sgRNA1-F	ATTGGTTGAGAACAGGGGTAACGG	
sgRNA1-R	AAACCCGTTACCCCTGTTCTCAAC	
sgRNA2-F	ATTGGGGGATCATTCAGAGCAGGT	
sgRNA2-R	AAACACCTGCTCTGAATGATCCCC	
Acos Tomato Seq F	CACTTTCCGATCGCTGAGGT	
Acos Tomato Seq R	AAGCAGTGGAGCAAGTCATTT	
*hptII-F*	CGAAAAGTTCGACAGCGTC	450 bp
*hptII-R*	GGTGTCGTCCATCACAGTTTG	
*SlActin-F*	AGGCACACAGGTGTTATGGT	186 bp
*SlActin-R*	AGCAACTCGAAGCTCATTGT	
SlACO2-qPCR-F	TCGCTGAGGTGGAGATACGGTGT	84 bp
SlACO2-qPCR-R	AACAGGGGTAACGGCGGCAG	
M13 F seq primer (−20)	GTAAAACGACGGCCAGT	
M13 R seq primer (−20)	GTTTTCCCAGTCACGAC	

## Data Availability

Not applicable.

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
