# Peer review of "Genome-Wide Identification of the Aconitase Gene Family in Tomato (*Solanum lycopersicum*) and CRISPR-Based Functional Characterization of *SlACO2* on Male-Sterility"

_ijms, 2022, doi:10.3390/ijms232213963_

Round 1

Reviewer 1 Report

To,

The Editor,

IJMS, MDPI,

Manuscript ID: ijms-2035388

 Subject: Submission of comments of the manuscript in “IJMS"

 Dear Editor IJMS, MDPI,

 Thank you very much for the invitation to consider a potential reviewer for the manuscript (ID: ijms-2035388). My comments responses are furnished below as per each reviewer’s comments. 

In the reviewed manuscript, the authors were was investigated Acotinase (SlACO) gene family in tomato using in-silico tools and functionally characterized with CRISPR/Cas9-mediated gene disruption. Genome-wide blast and HMM search represented two SlACO genes located on different tomato chromosomes. Both genes were estimated to have a segmental duplication in the tomato genome due to their identical motif and domain structure. One of these genes, SlACO2, showed a high expression profile in all generative cells of tomato. Therefore, the SlACO2 gene was targeted with two different gRNA/Cas9 construct to identify their functional role in tomato. The gene was mutated in a total of 6 genome-edited tomato lines, 2 of which were homozygous. Surprisingly, pollen viability was found to be extremely low in mutant plants compared to their wild type. Likewise, the number of seeds per fruit also sharply decreased more than fivefold in mutant lines (10-12 seed) compared to that in WT (67 seed). The pollen shape, anther structures, and flower colors/shapes were not significantly varied between the mutant and WT tomatoes. The mutated lines were also subjected to salt and mannitol-mediated drought stress to test the effect of SlACO2 on abiotic stress tolerance. The results of the study indicated that mutant tomatoes have higher tolerance with significantly lower MDA content under stress conditions. This is the first CRISPR-mediated characterization of ACO genes on pollen viability, seed formation, and abiotic stress tolerance in tomato.

  1. The subject of the manuscript is topical, very significant and interesting. The topic is interesting and well within the aims of the Journal. The study is well-conducted and provided important results that might use to develop a male sterile line and improve abiotic stress tolerance in tomato. However, some minor revisions are suggested as shown below;
  2. In general, a careful revision of the language must be carried out as well as of the punctuation. Moreover, several paragraphs are disconnected from each other and sometimes repetitive. Here some tips to improve the work, but as already said all the work needs a more accurate revision.
  3. the qRT-PCR methodology provided is also very vague and confusing. Please provide more details like the calibrator used in the study. I assume the authors have used the control as the calibrator. If so, the authors should not include the control within the bar graph as it represents the fold change between the treated vs control and a fold change of “1” for the ‘control’ doesn’t make any sense.  Also, would be good to provide details on what reagents (details of probes used, if any, if SYBR was used then details for that, etc.) and real-time PCR machines were used in the current study.
  4. References: shall have to correct the whole References according to the ”Instructions for the Authors”, e.g. title should not be in italics, the Journal name is in italics, and the author shall have to use the abbreviated name Journals cited the year must be bold. Please check all references carefully.

Author Response

Response to Reviewer 1 Comments

Thank you very much for your valuable suggestions. According to your proposal, we organized the whole manuscript, and we made the required corrections.

Point 1:  In general, a careful revision of the language must be carried out as well as of the punctuation. Moreover, several paragraphs are disconnected from each other and sometimes repetitive. Here some tips to improve the work, but as already said all the work needs a more accurate revision.

Response 1: The whole article was carefully re-read, especially the mistakes and flow problems in the paragraphs were corrected.

Point 2:  the qRT-PCR methodology provided is also very vague and confusing. Please provide more details like the calibrator used in the study. I assume the authors have used the control as the calibrator. If so, the authors should not include the control within the bar graph as it represents the fold change between the treated vs control and a fold change of “1” for the ‘control’ doesn’t make any sense.  Also, would be good to provide details on what reagents (details of probes used, if any, if SYBR was used then details for that, etc.) and real-time PCR machines were used in the current study.

Response 2: We used actin one gene as an internal reference gene. We changed Figure 6 and just show the relative expression of SlACOs in mutant lines compared to WT.

Point 3: References: shall have to correct the whole References according to the ”Instructions for the Authors”, e.g. title should not be in italics, the Journal name is in italics, and the author shall have to use the abbreviated name Journals cited the year must be bold. Please check all references carefully.

Response 3: We updated the reference list.

Reviewer 2 Report

The following research article entitled "Genome-wide identification of the aconitase gene family in tomato (Solanum lycopersicum) and CRISPR-based functional characterization of SlACO2 on male-sterility" focuses on the identification of aconitase gene family in tomato and the exploration of the putative role of SlACO2, one of the two identified ACO genes in tomato in seed formation and male sterility. This work presents a novelty, since it is, to my knowledge and as mentioned by the authors, the first report aiming the characterization of this gene family in tomato and the study of their function in pollen viability and seed establishment in tomato using a CRISPR mediated technology.

However, I am confused when it comes to the last section of the results related to the evaluation of the tolerance of SlACO2 lines to salt and osmotic stresses. In my opinion, the proline and MDA analysis at the stage of seedlings (10 days of culture) are not sufficient to strongly suggest the involvement of SlACO2 gene in tomato tolerance to both salt and osmotic stresses. More analysis should be conducted to support authors allegations. If not, the authors should remove the sentence related to abiotic stress tolerance from abstract, discussion and conclusion sections.

 Besides, below, some remarks and suggestions for the authors.

1.       Plants names should be written in italics. Make sure to write all the scientific name in italics. e.g: Arabidopsis thaliana, Solanum lycopersicum, Populus trichocarpa ….

2.       Genes abbreviations and names should be written in italics also. e.g. SlACO1, SlACO2

3.       More details should be added to figure captions to give more depth to the evaluated parameters. This is the case for example of the Figure 4 and 6. e.g. In figure 6, the analyzed gene name should me mentioned to avoid confusion.

4.       Include statistical analysis of the expression of SlACO2 in both WT and transgenic lines (Line 3 and Line 6). No letters had been added to the bars, which is in contradiction with figure caption.

Author Response

Response to Reviewer 2 Comments

Point 1: However, I am confused when it comes to the last section of the results related to the evaluation of the tolerance of SlACO2 lines to salt and osmotic stresses. In my opinion, the proline and MDA analysis at the stage of seedlings (10 days of culture) are not sufficient to strongly suggest the involvement of SlACO2 gene in tomato tolerance to both salt and osmotic stresses. More analysis should be conducted to support authors allegations. If not, the authors should remove the sentence related to abiotic stress tolerance from abstract, discussion and conclusion sections.

Response 1: Thank you very much for your valuable suggestions. We used 10-week-old plants for MDA and Proline assays and transferred these plants to new media every 3 weeks. In our study, we clearly saw the effect of stress applications at the end of this period. We collected leaf samples as soon as the plants became stressed and turned yellow. We think that we have sufficient experience on this subject, as we have had articles with these parameters before.

Point 2: Plants names should be written in italics. Make sure to write all the scientific name in italics. e.g: Arabidopsis thalianaSolanum lycopersicumPopulus trichocarpa ….

Response 2: All required corrections were made.

Point 3: Genes abbreviations and names should be written in italics also. e.g. SlACO1SlACO2

Response 3: All required corrections were made.

Point 4: More details should be added to figure captions to give more depth to the evaluated parameters. This is the case for example of the Figure 4 and 6. e.g. In figure 6, the analyzed gene name should me mentioned to avoid confusion.

Response 4: Figure captions were updated.

Point 5: Include statistical analysis of the expression of SlACO2 in both WT and transgenic lines (Line 3 and Line 6). No letters had been added to the bars, which is in contradiction with figure caption.

Response 5: Figure 6 was updated.